# Text Neck Syndrome in Children and Adolescents

**DOI:** 10.3390/ijerph18041565

**Published:** 2021-02-07

**Authors:** Daniela David, Cosimo Giannini, Francesco Chiarelli, Angelika Mohn

**Affiliations:** Department of Pediatrics, University of Chieti, 66100 Chieti, Italy; danieladavid88@gmail.com (D.D.); cosimogiannini@hotmail.it (C.G.); chiarelli@unich.it (F.C.)

**Keywords:** musculoskeletal pain, text neck syndrome, neck pain, mobile phone, children and adolescent health, technology addiction

## Abstract

Neck pain is a prevalent health problem, largely reported in adult patients. However, very recent data show that new technologies are inducing a shift in the prevalence of this relevant issue from adulthood to all of the pediatric ages. In fact, the precocious and inappropriate use of personal computers and especially cell phones might be related to the development of a complex cluster of clinical symptoms commonly defined as “text neck syndrome”. The purpose of this article is to analyze the new phenomenon of the “text neck syndrome”, the underlying causes and risk factors of musculoskeletal pain, that can be modified by changes in routine life, in different cultures and habits, and on the “text neck syndrome” as increased stresses on the cervical spine, that can lead to cervical degeneration along with other developmental, medical, psychological, and social complications. Findings support the contention that an appropriate approach for an early diagnosis and treatment is crucial to properly evaluate this emerging issue worldwide in children and adolescents who spend a lot of time watching smartphones and computers; additional research with more rigorous study designs and objective measures of musculoskeletal pain are needed to confirm significant relationships. Existing evidence is limited by non-objective measures and the subjective nature of musculoskeletal pain.

## 1. Case Report

A 16-year-old girl was admitted to a pediatric unit for a medical history of headache, dizziness, and acute neck pain. The referred symptomatology was headache, subjective vertigo, and ataxia. No fever or trauma were referred. Blood exams (complete blood count with formula, hepato-renal function, inflammation indexes, and beta-HCG levels) resulted in the normality range. Ophthalmologist examination resulted negative. Otorhinolaryngology’s examination excluded vestibular pathology. Medullary syndrome was considered by the neurologist.

An MRI (Magnetic resonance imaging) of the cervical spine was obtained. This exam showed an inversion of physiological cervical lordosis and posterior disc protrusion at C4–C5 level.

The hypothesis of text-neck syndrome was made. The girl was a high-school student, she described a daily routine of almost 6 h/day committed to study. 

The patient was re-evaluated by the orthopedic consultant and the physiatrist, who underlined the importance of primary prevention of musculoskeletal pain, avoiding excessive forward bending of the neck, thus maintaining ergonomic postures during study hours, and also limiting the use of mobile devices.

She was discharged with the indication not to watch touch-screen devices over 2 h/day and to help herself to study in a better position, paying attention to posture angles. 

This case highlights the importance of the new phenomenon of the text neck syndrome.

## 2. Definition

During the last few years, a growing reporting of data is showing that the “text neck syndrome” might be considered as an emerging 21st-century syndrome. This clinical condition refers to the onset of cervical spinal degeneration that results from the repeated stress of frequent forward head flexion while we look down at the screens of mobile devices and while we “text” for long periods of time [1,2]. Text neck syndrome is more common in adolescents, who, for several hours a day and for several days a year, hunch over smartphones and personal computers more frequently than in the past [3]. It is estimated that 75% of the world’s population is hunched over their handheld devices hours daily with their heads flexed forward [4].

### 2.1. History Background and Epidemiology

Neck pain is a very complex and important public health problem in our modern societies [5,6,7]. Any structure of the neck, such as intervertebral discs, ligaments, muscles, facet joints, dura, and nerve roots, might represent the origin site of the pain [8]. Several pathologies might be the cause of neck pain, such as tumors, infection, inflammatory diseases, and congenital disorders. However, in most cases no systemic illness is detected, thus resulting in a clinical condition named “musculoskeletal neck pain” [9].

In epidemiology studies evaluating the general population, the 1-year incidence of neck pain can be as high as 40% [10]. Among all health conditions for years lived with disability, the World Health Organization (WHO) has classified neck pain and other musculoskeletal diseases as the 4th and the 10th pathological condition, respectively [11]. As well, this report showed that these conditions represent the key drivers of the increase in years lived with disability over the past 20 years [11]. Similar data have been reported in childhood. In fact, according to the WHO Global Burden of Disease, neck pain is the 8th ranked reason for most years lived with disability for 15–19 year olds of any health condition, which is higher than other well-known adolescent health problems, such as asthma, alcohol use, drug use, and road injury [12].

In adults, there are extensive data about the epidemiology, burden and treatment of musculoskeletal pain, but contrasting and not universally accepted results are reported in children and adolescents. In fact, the lack of clinical research in children and adolescents has been emphasized by several studies [13,14,15,16,17]. Emerging evidence shows that children and adolescents with persistent pain are at an increased risk of chronic pain as adults [4,18,19]. Moreover, a lot of musculoskeletal illnesses follow a pattern of long-term recurring exacerbations and remissions and the better predictor of a new episode is the experience of a previous episode [20].

As we know that the prevalence of musculoskeletal conditions in childhood and adolescence is always increasing, it may be important to investigate the condition early in life and to understand the main aspects and risk factors of the onset of the symptoms, in the way to provide and to develop the best and the most efficacious treatments [21]. In addition, a complete understanding of the disease might help to adopt all the efforts for primary prevention of this condition [1].

### 2.2. Pathology

The weight of the head on the spine is dramatically increased when it is flexed forward, and the effects and amount of weight are strongly and progressively enhanced by varying the degrees (Figure 1). In fact, a full-grown head weights almost 5 kg in the neutral position [5]. The more the head is flexed, more the forces on the neck surge to more than the double at 15° (roughly 12 kg). In addition, the burden of the weight of the head increases to 18.14 kg at 30° and to 22.23 kg at 45°, reaching a more than fivefold effect at 60°, arriving to a 27.22 kg [5]. Not only the degree of the neck flexion is relevant but also the frequency of head banding induces additive effects on the neck physiology. In fact, the frequent forward flexion can change the cervical spine, curvature, supporting ligaments, tendons, musculature, the bony segments, commonly causing postural change and pain on the neck and associated areas [2].

It is estimated that children and adolescents spend a medium of 5 to 7 h a day on their smartphones and handheld devices with their heads flexed forward to read and text. It has been reported that the cumulative effects of this exposure reach alarming results of an excess stress on the cervical spine area, ranging from an average of 1825 to 2555 h a year [5]. Therefore, it is very important for clinicians and health care providers to evaluate and properly characterize the condition in order to adopt all the procedures for the screening of subject at increased risk as well as for timely diagnosis and treatment in childhood.

### 2.3. Clinical Features

There are several complications of text-neck syndrome. They can involve the eyes, the heart and lungs, the head, and the psychological field. An elevated number of studies all over the world have analyzed musculoskeletal pain in children and adolescents. There is now enough evidence to support the association between flexing toward the neck and symptoms referred to the cervical spine.

Children and adolescents do not take the long-term damage to the body seriously or do not know about it, probably because the short-term effects are not so noticeable. Only in adulthood can the effects of forward flexion of the neck seriously affect the quality of life. This fact must put the attention on younger people, the most frequent users of smartphones and tablets: this increase the fear that young people could face a future of pain and disability, or even worse taking years off of their life expectancy [1].

#### 2.3.1. Musculoskeletal Pain

Some authors conducted a study in which they examined a sample of 207 children and adolescents with nonspecific neck pain [1]. In all patients (100% of the sample), a cervical neck pain irradiated to the back and the shoulders was reported. This pain had duration of more than 6 months and was not associated to sensory or motor deficits. Twenty-seven patients with dorso-lumbar scoliosis were excluded from the study; the study focused on the remaining 180 patients, without any other diseases, but with the diagnosis of musculoskeletal neck pain with spasm. The age range was from 8 to 17 years, with the mean age of 14 years. The demographic characteristics of the children and adolescents in the study show that all the 180 participants (100%) reported forward flexion of neck while they are studying and while they are using smartphones and/or tablets: the study demonstrates that all participants (100%) presented a strong flexion of the neck (≥45 degrees) during the daily activities and that all participants spent an average of 5 to 7 h a day on their smartphones and handheld devices. As side-effects on short-time, the study shows that the main pain location is on the neck (100% of the sample), followed by pain on shoulders (69%), lower back (61%), and then arms (13%); the eye symptoms are eye strain (12%), dry eyes (7%), and near sightedness (3%); the psychological and social effects evidenced in this study are irritability (82%), stress (62%), anxiety (59%), poor communication (82%), and decreased school marks (64%) [1].

Another study evaluated upper quadrant musculoskeletal pain (UQMP) in children and adolescents as a common health problem, due to the increased sedentary lifestyles and the growing use of screen-based activities [22,23]. The aim of this study was to understand the risk factors to implement preventive and treatment strategies. It has been described how different sitting postures can influence the head/neck posture and the activity of the muscles of the cervicothoracic spine in a very significant way [24]. Therefore, the sitting position and the duration can contribute to low back pain and especially on the UQMP. An evaluation questionnaire about the causative factors for UQMP described by children and adolescents was put to a group of children and adolescents, to determine if there is evidence of correlation between sitting and UQMP and the different elements of sitting related to UQMP in this population. The aim of the study was to understand the assessment of posture that provides information about the biomechanical alignment of the bone structures at any specific moment in time. Four elements related to UQMP were identified: the sitting duration [25,26]; the activities while sitting [25,26,27,28]; the dynamism (amount of movement while seated) [26,29]; and postural angles (spinal angles while seated) [30,31,32,33,34]. It was evaluated that, if sitting posture is prolonged and static, certain anatomical structures are adversely affected by prolonged strain; time by time these structures could consequently become the cause of musculoskeletal pain [35,36]. So, it can be concluded that there is an unequivocal link between sitting and UQMP in children and adolescents and that postural angles during sitting should be considered a possible risk factor for UQMP, that could be better explored in future research.

In other studies, a significant association between sitting and UQMP in children and adolescents has been highlighted [26,27,28,29,30,31,32,33,36,37]. Another study evaluated a cohort of asymptomatic high-school students, with the aim to describe the variability of five postural while working on desktop computers and the relationship between the postural angles and age, gender, height, weight, and computer use; 821 students were screened at baseline and 240 students after one year of follow-up [38,39].

Straker et al. studied also the postural differences between adolescent computer and non-computer users and found that computer users had increased neck flexion and increased pelvic tilt [33]. The same authors also showed that the increased computer use was associated to increased head flexion and neck flexion especially for boys and increased lumbar lordosis for girls [40].

In an Australian study, 33 adolescents aged from 12 to 15 years were evaluated to verify the correlation between information and communication technology (ICT) use and musculoskeletal discomfort. The participants were eligible if they used ICT at least once per week, for almost 15 min at a time, in their normal activities; adolescents with a pre-existing diagnosis of musculoskeletal disorder were excluded. This study clearly showed an increase of prevalence of low-level discomfort, more frequently in the legs, head/neck, back and shoulders in those using ICT. The types of ICT most used were TV, desktop and laptop computers, mobile telephones, and portable hand-held gaming devices. The specific areas involved with ICT use, such as head, neck, shoulders, and upper back and arms/hand were significantly associated with the amount of ICT use and discomfort described at the end of the day. No significant correlation was found between the amount of ICT use and the discomfort in the areas of the body commonly affected by ICT use (such the head, neck, shoulders, upper back and arms/hands) [41].

A study in Hong Kong examined the ergonomic issues involved in computer workstations: the study was conducted in the home environment of students at primary school age. The authors considered different factors, such as anthropometric, postural, and ergonomic considerations, to show the effect of computer use in the home environment on the children. A sample of 15 participants, 6 males and 9 females, aged from 8 to 11 years old, all attending the primary school levels 3–6 were collected. This sample did not highlight discomfort symptoms related to computer use, maybe because they were not intensive users or because they do not suffer serious musculoskeletal symptoms related to computer use. Only 4 out of 15 highlighted a discomfort directly related to computer use, while in other studies with a larger sample the discomfort was more frequently evaluated. This study aimed to investigate how computer use in primary school students in the home environment can influence musculoskeletal symptoms. It was identified that the computer furniture is not ideal for users; this contributes to have constrained and awkward postures during computer use. It is possible that these mal-adaptive postures may become habitual and extended into adolescence or adulthood. About 20–30% of the children reported musculoskeletal discomfort related to computer use. It becomes very important to understand the ergonomic issues in more extensive research studies, maybe using motion analysis and electromyography measurements to better understand children’s postural control and variability during computer use in the home environment [42].

Another study examined the adolescent upper quadrant musculoskeletal pain (UQMP), as a significant health concern with a worldwide prevalence of 30% [43,44,45]. Also in this study, the discomfort for musculoskeletal pain in adolescents is related to several causes: reduced social interactions, mental health, school attendance, scholastic competence, and participation in physical activities [44,46]. The etiology of adolescent UQMP is multifactorial, it can include complex physical and psychosocial factors [44,47,48]. Nowadays, adolescents are increasingly exposed to screen-based activities at home and also at school [23].

A large number of studies report a significant association in high school students between neck pain and weekly computer use of nine or more hours [49]. The aim of one study was to identify specific spinal segmental postures associated with development of UQMP in a 12-month period. A sample of 240 high-school students was evaluated, aged between 15 and 17 years, living in the Western Cape metropole (South Africa), asymptomatic and naïve to computing classes. The students were screened for UQMP by completing the pain section of the Computer Usage Questionnaire (CUQ), developed and validated in South African high school settings [50]. In this study, for the first time, the development of UQMP is described in relation to postural angles, computer use, anxiety, and depression. This study also analyzed the correlation between increased head flexion (HF), defined as the head-on-neck alignment and seated-related adolescent UQMP, developed in a period of 6 to 12 months, for students that use computing studies at school [30]. Thus an association between UQMP and computer use has been demonstrated [51]; in addition, prolonged sitting postures could be the cause of musculoskeletal disorders on head, neck, shoulder, and midback pain [25,27,36]. In 44% of cases, headache and neck pain could be related to pathological changes in the upper cervical structures in response to increased load on active and passive structures, because of the upper cervical nerve innervation [52,53,54]. The most important anatomical structure is the trapezius muscle, which is fundamental in the linking between the spinal column and the upper arm [55]. This muscle is innervated by the spinal accessory nerve (Cranial nerve XI) and the cervical plexus (C1-C4) [56]. The increased HF could potentially lead to pain in these nerves; this has been confirmed in a study in which the authors reported a positive correlation between upper trapezius muscle activity and HF in children [57]. Further research must investigate the field of classroom furniture (chair, desk, and monitor height) and postural hygiene (knowledge and postural habits), which could contribute to increased HF posture.

In addition, a study performed in Thailand investigated the common complaint of non-specific symptoms of musculoskeletal pain that can have a sudden onset in adolescent age and can have long-lasting symptoms [18,58]. The symptoms related to musculoskeletal pain that affect a large portion of school-adolescents is a great health burden and an increased cause of augmented cost living because it involve the performance of daily living activities such as studying, exercising, or social participation. Musculoskeletal pain at the shoulders is found to be associated with headache, that is one of the most common public health problems in children and adolescents; it can cause muscle tension and it can be attributed also to tight muscles in the shoulders, neck, scalp, and jaw [59,60].

A cross-sectional questionnaire was used in primary and secondary Thailand schools, in Khon Kaen and Phitsanulok Provinces, from November 2009 to June 2011, recruiting 2750 participants. Thai school-age adolescents reported a high prevalence of neck and shoulder pain, as European adolescents (from 15% to 28%) [61,62] and Chinese teen-agers (41.1%) [63].

The prevalence of musculoskeletal pain symptoms in school-age adolescents is different among age, groups, and sex. Headache was the most common musculoskeletal pain symptom referred with the highest prevalence in Thai school-age adolescents. Additionally, it was found that headache and musculoskeletal pain symptoms were more prevalent in girls than in boys and these rates tend to be higher in older adolescents. Ankle or foot pain was a common musculoskeletal pain among younger boys and girls playing sports. Neck and shoulder pain were a common musculoskeletal pain among older boys and girls and were associated with computer use and school bag carrying.

Chronic pain in the neck and shoulders can be referred to the head, causing headache, especially tension-type headache, which is quite common in adolescents. Trigger points in the head, neck, and shoulders shared similar pain patterns with chronic tension type of headache in children [64], so we can justify why neck and shoulder pain can begin in early adolescence and persist into chronic musculoskeletal problems in adulthood [58].

#### 2.3.2. Eye Symptoms

The effects of texting with the neck in forward flexion can cause nearsightedness, eye strain, or dry eyes, because the focus on the object is nearby [65]. In a recent study the most frequent eye symptoms were eye strain (12%), dry eyes (7%), and near sightedness (3%) [1].

#### 2.3.3. Electromagnetic Risk

Another complication of texting is electromagnetic radiation: according to a recent study, cell phones, computers, wireless internet, and televisions generate an extremely low-frequency electromagnetic field [66]. Electromagnetic radiation can cause a wide variety of symptoms: difficult sleeping, dizziness, headaches, tingling in the hands, ringing in the ears, eye pain, cardiac conditions that can’t be explained, electro sensitivity, low immunity, attention deficit hyperactivity disorder, and autism [67]. This is very important because the absorption of electromagnetic radiation on a child’s head can be greater by over two times, and absorption of the skull’s bone marrow can be greater by over ten times than in adults [68].

#### 2.3.4. Psychological Effect

Changes in the behavior of children and adolescents in daily habits and usual social interactions, noticed by parents, have also been reported; these symptoms have been reported in 147 young participants (82%). Parents described them as more irritable and alienated, and parents of 115 participants (64%) reported a decline of the school grades of their sons [1].

The psychological sphere is strongly involved: pediatricians must pay attention to what parents are worried about. In a study by Emre et al., parents report that most of the children/adolescents who participated in the study reported a high grade of isolation and easy irritability [66]. Their grades in school were negatively affected. Children who use more than 1–2 h per day of technology (the limit that experts recommend) have an increase of almost 60% in psychological disorders [69]. We know that children and adolescents spent averages of 5 and 7 h a day, respectively, on their smartphones and handheld devices. Therefore, it is important to underline that the more time students spend consuming media, and the more violent its contents are, the worse their grades in school are, even when family, education, or immigrant background are controlled for [70]. In addition, it has been noticed that spending too many hours on handheld devices and smartphones negatively affect children and adolescents’ communication skills, particularly in face-to-face communication skills, bullying, and teasing [71].

#### 2.3.5. Further Comorbidities

Some new studies have suggested that forward postures, like we have while studying, emailing, texting, surfing the web, and playing video games, are related to hyper-kyphosis, which is associated to cardiovascular problems and pulmonary disease. In fact, when someone looks at a smartphone or a tablet, he drops his head and rounds his shoulders while looking down, so there is a restriction of the muscles of the ribs and impeachment in movement that make it harder to take a full breath [72].

There is also an important relation between increased neck flexion and increased weight, probably due to the decreased physical activity [40].

The prevalence of headaches, especially migraines, has been studied among school-age children and adolescents, and it varies according to age and sex [73]. A Swedish study found a 48% incidence of headache among schoolchildren aged 7–18 years, while Fichtel and Larsson found that headache is the most common pain among girls (42%) [74,75].

A study conducted by the Commonsense Census analyzed the mobile media and the use in children and adolescents. It demonstrated that mobile media have become an essential part of the children’s media landscape, across all levels of society.

Nearly the totality of children (98%) (aged 8 or less) have some type of mobile device and a TV in their home. Ninety-five percent of families with children have a smartphone, with a trending up from 41% in 2011 to 63% in 2013, and 78% of families have a tablet (from 8% in 2011 up to 40% in 2013). Indeed, 42% of children have their own tablet device, from less than 1% in 2011 up to 7% in 2013. The main activities of children on mobile devices are watching TV and playing video games. It is estimated that around 7 out of 10 children under the age of 8 (around 70%) have watched videos on YouTube in their lifetime and played video games on a mobile device, while only 59% of children have watched TV shows or movies on a mobile device. Only 28% of children have read a book on a smartphone or tablet [76].

## 3. Prevention

Musculoskeletal neck pain is a multifactorial disease that has become common in children and adolescents. Numerous risk factors are known to contribute to the development of this pathology. Movements such as bending the head, neck, and shoulders on cell phones and portable devices, and distorting the neck position while sitting, studying, or watching television, can progressively increase stress in the cervical spine. This stress can result in various complications, such as premature wear, lacerations, degeneration, possible surgery, developmental, medical, psychological, and social complications. While it is very difficult to avoid habits in bad postures and avoid overuse of modern technologies, today’s young people should strive to perform their daily activities while keeping their spine in a neutral position as much as possible, avoiding excessive neck flexion for several hours every day [7,77,78,79].

Various indicators of psycho-physical well-being in children have been negatively associated with insufficient physical activity [80,81], and excessive sedentary time [82,83], particularly in front of the screen [84]. According to the latest evidence, fewer than 10% of children in Canada now meet the current guideline of 60 min of moderate to vigorous physical activity (MVPA) per day [85].

According to data recently published by the WHO, overweight and obesity are among the most serious public health problems of the 21st century. The increasing prevalence of excessive weight in childhood can consequently cause an increase in the problems of suboptimal posture in children and adolescents [86]. The increase in adipose tissue influences the onset of postural defects found above all in the region of the shoulders and pelvis, as a study of Rusek et al. demonstrated [87], in which a higher content of fatty tissue was associated with greater asymmetry in the scapular area. It is obvious that the school environment, where children spend about 1/3 of their time (on average 6–8 h a day), deeply influences their psycho-physical development. It should be noted that in addition to the hours spent sitting at school desks, children spend more and more time sitting in front of the TV and computer, to the detriment of the hours spent on the move, and this can only exacerbate the occurrence of wrong postures and the weight gain [86].

It has been shown that the prevalence of neck pain is higher in older adolescents and that physical inactivity and professional activities can be a risk factor for the onset of back pain [88].

In fact, in a three-year longitudinal study by Wedderkopp et al., physically active adolescents were less likely to develop back pain [89]. Physical inactivity has been associated with an increase in sedentary behavior, such as computer and television use [90], and this high stationary period may contribute to changes in posture and subsequently back pain [91], although it is necessary to consider in young people the care related to posture doing physical activity [88]. 

Many information campaigns have aimed to raise awareness of the problems related to musculoskeletal pain (neck pain in particular), with particular regard to the possible medical, psychological, and social consequences and the methods of prevention and treatment [1].

Prevention is the key when it comes to text-neck. The following suggestions should be kept in mind while using smartphones or other handheld devices [92]:Avoid excessive usage and take frequent breaks.Avoid prolonged static postures.Position the device such that it reduces stresses both on the head/neck and the upper extremities.Avoid high repetitions of movements such as prolonged typing or swiping.Avoid holding large or heavy devices in one hand for long duration.

The Australian Department of Health (2012) published guidelines for children about the recommended time to pass on mobile devices or electronic media [93,94]:Children < 2 years: recommended no time watching TV or using other electronic media (DVDs, computer, and other electronic games).Children 2–5 years: no more than 1 h/day sitting and watching television and other electronic media (DVDs, computer, and electronic games).Infants, toddlers, and preschoolers should not be sedentary, restrained or kept inactive for more than 1 h at a time, with the exception of sleeping.

Even the Italian Society of Pediatrics released their top-5 rules for families with children in 2019. These are the “*Top 5 recommendations of Italian Society of Pediatrics (2019)*” [95]:

**1. Talk with your son**—It is important to encourage open communication between parents and adolescents, explaining to children what a positive and intelligent use of media devices means, paying attention to the contents that are published and read and reminding them that it is essential to protect online privacy to protect themselves and your family.

**2. Understand, learn and control**—Parents should monitor the time their child spends on tablets, smartphones, and PCs, first learning the available technologies to be able to understand them as much as possible, and playing with them and sharing activities on media devices as much as possible.

**3. Set clear limits and rules**—It is necessary to limit the time of use of smartphones, tablets and PCs during the day or on weekends, by establishing precise times of prohibition, for example during meals, homework, and family gatherings. Consider media an opportunity for the whole family to watch movies together or share social content or use chat and video messages.

**4. Give a good example**—The parent’s example is essential, so fathers and mothers should be the first to limit the use of smartphones when they are with the family and during meals. It is also important that the parents choose appropriate content and languages on social networks.

**5. Do network**—Collaboration between parents, pediatricians, and health professionals is essential to protect and support children through information campaigns that provide greater awareness of the positive aspects but also of the risks of excessive use of media devices.

## 4. Conclusions

The aim of this article was to examine the problem of musculoskeletal pain in children and adolescents, since it is a common multifactorial disease. It is known that there are numerous potentially modifiable risk factors contributing to its development. The increased stress on the cervical spine is due to the continuous flexion movements of the head and neck on mobile devices used daily.

Childhood and adolescence are characterized by periods of rapid growth and many physical problems can occur due to prolonged and excessive use of smartphones or computers, although it is often thought that this type of problem is more common in adults. Recently, the literature has focused attention on issues that can become a serious health problem, such as the increase in sedentary living and the excessive use of electronic devices among young people.

All of the studies carried out on musculoskeletal pain in the pediatric age show the same characteristics: the specific symptoms (headache and musculoskeletal pain) are expressed more in females than in males, and these rates are higher in adolescents than in children. Joint pains, especially in the lower limbs (ankle or foot), are more common among younger boys and girls who play sports. On the other hand, musculoskeletal pain referred to the cervical spine is more frequent among older boys and girls, often associated with carrying a backpack and using a computer for study purposes.

Symptoms of chronic pain to the cervical spine and shoulders can sometimes also be referred to the head, causing headache and muscle tension, frequently encountered in adolescents. In fact, the trigger points of the nociceptive signal in the head, neck and shoulders share the same pathways as chronic tension headache in children, so this may justify why neck and shoulder pain can begin in childhood and early adolescence and persist into adulthood with chronic musculoskeletal problems. Thus, maintaining incorrect sitting postures for a long time could cause the appearance and persistence of musculoskeletal disorders related to the head, neck, shoulders, and dorsal-lumbo-sacral spine.

Currently, studies on musculoskeletal pain in children and adolescents are still limited, so there is a lack of validated tools to have an objective measure of musculoskeletal cervical pain and to evaluate the direct consequences. Therefore, it is difficult to draw conclusions based only on self-administered tools, such as questionnaires. Furthermore, there is the difficulty of comparing different samples, populations and cultures to consider. Future studies should therefore aim to validate a unified tool for measuring musculoskeletal pain in the neck and make it comparable in different cultures. It is known that pain experienced in childhood is related to pain reported in adulthood, although the pathophysiological mechanisms are still unknown; several studies have hypothesized that in cases of psychological triggers in childhood (such as a particular painful experience), in genetically predisposed subjects, there is also a predisposition to develop chronic pain in adulthood. Further studies would be needed in the future to explore and investigate the mechanisms responsible for these associations.

In addition, further research is needed to improve school furniture in classrooms (chair, desk and monitor height), and about postural hygiene (knowledge and postural habits), to help improving the head flexion (HF) posture.

Although it is very difficult not to use modern technologies or to avoid incorrect postural habits, the cause of musculoskeletal problems, information policies should be implemented for young people to be aware of the most correct postures. For their part, young people should strive to perform their activities by keeping the spine neutral and avoiding excessive neck flexion for hours on end every day.

The problem will probably worsen in the near future and will be even more relevant, in light of the COVID-19 pandemic; in fact, distance learning made inevitable forces an entire generations of children and adolescents to spend even more time at home on books and inevitably on mobile devices.
**What is Already Known on This Subject?**➢Musculoskeletal pain is a predominant problem during childhood and adolescence.➢Low back and neck pain are exacerbated by the augmentation on forward flexion on mobile devices used daily.➢Recently, the literature has focused attention on the increase in sedentary living and the excessive use of electronic devices among young people that can become a serious health problem.➢Studies on musculoskeletal pain in children and adolescents are limited, because of the lack of validated tools to have an objective measure of musculoskeletal cervical pain and to evaluate the direct consequences. Therefore, it is difficult to draw conclusions based only on self-administered tools such as questionnaires. Furthermore, there is the difficulty of comparing different samples, populations, and cultures to consider.**What Does This Study Add?**➢This study examined current literature focusing on musculoskeletal pain in children and adolescents and to evaluate the direct consequences.➢It presents a comparison among different cultures and highlights the physio-pathological base of the pain experienced in childhood; probably in cases of psychological triggers in childhood (such as a particular painful experience), in genetically predisposed subjects, there is also a predisposition to develop chronic pain in adulthood.➢The study highlights the need for future studies to therefore aim to validate a unified tool for measuring musculoskeletal pain in the neck and make it comparable in different cultures and to investigate the mechanisms responsible for the associations in genetically predisposed subjects to develop chronic pain in adulthood.➢This study concludes that further research is needed to improve school furniture in the classrooms (chair, desk, and monitor height) and about postural hygiene (knowledge and postural habits), to help improving the head flexion (HF) posture.➢The COVID-19 pandemic will probably exacerbate the problem.

## Figures and Tables

**Figure 1 ijerph-18-01565-f001:**
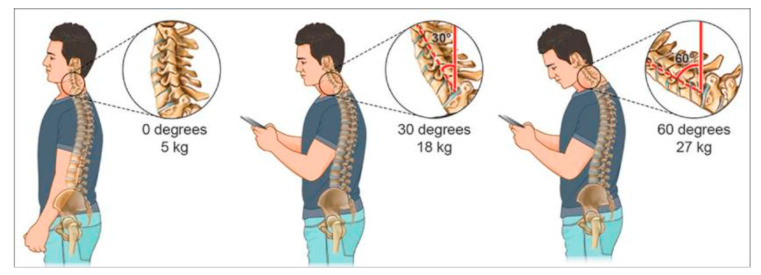
A chart depicting the stress and weight put on the neck and spine as a result of hunching over a smartphone and handheld devices at varying degrees. The neck flexion angle is the angle between the global vertical and the vector pointing from C7 to the occipitocervical joint. A fullgrown head weighs 5 kg in the neutral position. As the head bends forward, the weight seen by the neck increases to 18 kg at 30° and 27 kg at 60°. Reprinted with permission from ref. [1]. Copyright 2017 Surgical Neurology International.

## Data Availability

Data sharing not applicable.

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
