# Peer review of "Text Neck Syndrome in Children and Adolescents"

_ijerph, 2021, doi:10.3390/ijerph18041565_

Round 1
Reviewer 1 Report
The paper at hand presents a case study about the "text neck syndrome" observed in a 16 year old girl. The paper is well written and easy to follow. The authors provide a quick overview about the actual case, and provide a comprehensive, literature based definition of the syndrome.
Additionally, a section about prevention of the text neck syndrome is added to the paper, before the authors finish the paper with a coherent conclusion.
The only issue I found: In line 35 the first name of the girl is mentioned, while the remaining paper talks about “the girl” only. To ensure anonymity of the subject the name should be removed.
Author Response
We are grateful to the Reviewer for peer-reviewing our manuscript, and for the valuable and constructive feedback provided.Reviewer 2 Report
The Authors of the paper "Text Neck Syndrome in Children and Adolescents" put a lot of effort into writing it. Despite the very interesting topic of the work, I have a few questions:
1. CASE REPORT section:
- Do the authors of the paper have information about the patient's childhood health problems? Was there any problem during the delivery. What was the baby's birth weight and body length washed. Since the patient started to have a headache.
- Has the patient consented to participate in the study and was she informed about the scientific publication?
- did the patient's parents consent to the scientific publication?
- Do the authors have the consent of the research ethics committee? If in Italy the consent of the bioethics commission is required, please send the relevant documentation, confirmed by the commission, or the regulation of the bioethics commission which states that CASE REPORTS do not require the consent of the commission.
- All this information should be included in the text.
2. PATHOLOGY section, it is also worth considering and adding figure 2 what the position of the child sitting incorrectly in front of the computer looks like (during learning or games).
3. Section Musculoskeletal pain and PREVENTION:
- Do the Authors provide information in the manuscript about the body posture when using phones and computers?
- I want to add a paragraph about body posture, how does it change with age and what posture errors occur in school-age children
- it is also worth adding a paragraph regarding the sedentary lifestyle of children as well as about reduced physical activity
Literature proposal:
- Scarabottolo CC, Pinto RZ, Oliveira CB, Zanuto EF, Cardoso JR, Christofaro DGD. Back and neck pain prevalence and their association with physical inactivity domains in adolescents. Eur Spine J. 2017 Sep;26(9):2274-2280. doi: 10.1007/s00586-017-5144-1. Epub 2017 May 23. PMID: 28536945.
- Rusek W, Baran J, Leszczak J, Adamczyk M, Weres A, Baran R, Inglot G, Pop T. The Influence of Body Mass Composition on the Postural Characterization of School-Age Children and Adolescents. Biomed Res Int. 2018 Oct 14;2018:9459014. doi: 10.1155/2018/9459014. PMID: 30406147; PMCID: PMC6204177.
- Garriguet D, Colley R, Bushnik T. Parent-Child association in physical activity and sedentary behaviour. Health Rep. 2017 Jun 21;28(6):3-11. PMID: 28636068.
References section:
Authors should carefully check the Instructions for Authors tab: (https://www.mdpi.com/journal/ijerph/instructions)
and adapt the literature requirements to the manuscript text:
References: References must be numbered in order of appearance in the text (including table captions and figure legends) and listed individually at the end of the manuscript. We recommend preparing the references with a bibliography software package, such as EndNote, Reference Manager or Zotero to avoid typing mistakes and duplicated references. We encourage citations to data, computer code and other citable research material. If available online, you may use reference style 9. below.
Citations and References in Supplementary files are permitted provided that they also appear in the main text and in the reference list.
In the text, reference numbers should be placed in square brackets [], and placed before the punctuation; for example [1], [1-3] or [1,3]. For embedded citations in the text with pagination, use both parentheses and brackets to indicate the reference number and page numbers; for example [5] (p. 10). or [6] (pp. 101–105).
The reference list should include the full title, as recommended by the ACS style guide. Style files for Endnote and Zotero are available.
References should be described as follows, depending on the type of work:
Journal Articles:
1. Author 1, A.B .; Author 2, C.D. Title of the article. Abbreviated Journal Name Year, Volume, page range.
Books and Book Chapters:
2. Author 1, A .; Author 2, B. Book Title, 3rd ed .; Publisher: Publisher Location, Country, Year; pp. 154–196.
3. Author 1, A .; Author 2, B. Title of the chapter. In Book Title, 2nd ed .; Editor 1, A., Editor 2, B., Eds .; Publisher: Publisher Location, Country, Year; Volume 3, pp. 154–196.
Unpublished work, submitted work, personal communication:
4. Author 1, A.B .; Author 2, C. Title of Unpublished Work. s
- tatus (unpublished; manuscript in preparation).
5. Author 1, A.B.; Author 2, C. Title of Unpublished Work. Abbreviated Journal Name stage of publication (under review; accepted; in press).
6. Author 1, A.B. (University, City, State, Country); Author 2, C. (Institute, City, State, Country). Personal communication, Year. - Conference Proceedings:
7. Author 1, A.B.; Author 2, C.D.; Author 3, E.F. Title of Presentation. In Title of the Collected Work (if available), Proceedings of the Name of the Conference, Location of Conference, Country, Date of Conference; Editor 1, Editor 2, Eds. (if available); Publisher: City, Country, Year (if available); Abstract Number (optional), Pagination (optional). - Thesis:
8. Author 1, A.B. Title of Thesis. Level of Thesis, Degree-Granting University, Location of University, Date of Completion. - Websites:
9. Title of Site. Available online: URL (accessed on Day Month Year).
Unlike published works, websites may change over time or disappear, so we encourage you create an archive of the cited website using a service such as WebCite. Archived websites should be cited using the link provided as follows:
10. Title of Site. URL (archived on Day Month Year).
See the Reference List and Citations Guide for more detailed information.
Author Response
We thank the Reviewer for peer-reviewing our manuscript and for those relevant observations.
CASE REPORT: The girl has never had any health problems. She was born at term, the neonatal period was normal. Headache was described shortly before admission.
The patient consented and her parents were informed about the scientific publication and they consented it.
PREVENTION section: In the manuscript, we wanted to emphasize the importance of correct posture and the effects that derive from wrong postures through the image in the text which is very explanatory. As the reviewer suggest, we added a paragraph about body posture and the sedentary life and reduced physical activity (lines 314-335).
Reviewer 3 Report
Dear Authors,
The manuscript has profound formal presentation errors.
It is essential that before sending your manuscript for evaluation, you carefully read the instructions for the authors (and this is especially serious in this case, given that the IJERPH instructions are very few and very easy to execute).
On the other hand, the study pathology (although recent) as indicated by the authors in the manuscript is relatively prevalent in today's society, so the publication of a case study of this nature would not be justified. At least in a journal of the scientific impact of the IJERPH.
Kind regards.
Author Response
We thank the Reviewer for the peer review and for this relevant observation. Our purpose with this article was to discuss an important and prevalent public health problem, starting with a clinical case that occurred in our department, and comparing interesting works carried out in different parts of the world, as we clarified in the key clinical message, of the abstract, as follows:
- “The purpose of this article is to analyze the new phenomenon of the “Text Neck Syndrome”, the underline causes and risk factors of musculoskeletal pain, that can be modified by changes in routine life, in different cultures and habits.”
Reviewer 4 Report
The article raises an important issue of health hygiene among children and adolescents. For obvious reasons, this subject is very important. However, in my opinion, the article is more suited to a popular science journal and does not bring much scientific knowledge to the topic. Therefore, I propose the rejection of this article.
As far as text-neck syndrome is concerned, it is not a separate disease, but it will probably happen over time if research shows that the pain is caused only by excessive use of electronic devices. However, the pain is most likely caused by the overlapping of multiple risk factors, of which the use of electronic devices is only one component.
First of all, the article is classified as a case report, and later the authors refer to it as a review. Second, as the authors noted, there is little research on the subject so far. Therefore, creating reviews describing the syndrome itself does not seem justified. And this is probably why the authors mix text neck syndrome with ordinary cervical spine pain in the text. In the abstract, the aim of the work is described as "analyze the new phenomenon of the" Text Neck Syndrome ", but in the conclusion as" examine the problem of musculoskeletal pain in children and adolescents ". Thirdly, the cited case report is too laconic, and it also describes that Elena spends 6 hours a day studying and does not describe how much time she spends with electronic devices. I would suggest carrying out a case series study, detailing the time spent with a smartphone and books separately.
Author Response
We thank the Reviewer for this relevant observation.
The article was built starting from a case report that is enriched with a mini-review of the literature, with the aim of giving a more complete and multi-ethnic framework to the certainly relevant problem.
Certainly, the text neck syndrome constitutes a pathological "continuum" with the most general musculoskeletal pains, but precisely, because of its high prevalence, it appeared important to mention the syndrome individually, but obviously placing it in the more generic category to which it belongs.
The case described is, certainly, short, because it is a patient with a mute medical history, who has always been well. Reference is made to books only because, as specified in the text, the problem is the excessive neck flexion, regardless of whether it is books or cell phones.
We also added a personal consideration in the final conclusion: “Probably the problem will worsen in the near future in light of the COVID-19 pandemic and the distance learning made inevitable”.
Round 2
Reviewer 2 Report
The authors adapted to the comments of the reviewer and made appropriate corrections in the text.
Author Response
We are grateful to the Reviewer for peer-reviewing our manuscript, and for the valuable and constructive feedback provided.Reviewer 3 Report
Dear Authors:
The high prevalence of the syndrome under study and the high availability of the resources used make it impossible to justify that the methodology used is a case report.
I encourage the authors to conduct research with experimental methodology on a larger sample size. In this way they will be able to obtain relevant results for the scientific community.
Kind regards.
Author Response
We thank the reviewer for the suggestion. We decided to use the clinical case that happened to our department as a starting point for a review of the literature on this topic. At the moment we have no further clinical cases in the ward to build a case series. It will be our concern in the future to establish a case series if we have the possibility.
The research design may not be appropriate as our article is a case report aimed at reviewing the literature but does not include a research design.